# Knowledge, acceptance and perception on COVID-19 vaccine among Malaysians: A web-based survey

**Nurul Azmawati Mohamed**, **Hana Maizuliana Solehan** *, **Mohd Dzulkhairi Mohd Rani**, **Muslimah Ithnin**, **Che Ilina Che Isahak**

Faculty of Medicine and Health Sciences, Universiti Sains Islam Malaysia, Nilai, Negeri Sembilan, Malaysia

☯ These authors contributed equally to this work.
* drhana@usim.edu.my

**Data Availability Statement:** The Supporting Information File is available at 10.6084/m9.figshare.14932605.

**Funding:** The authors received no specific funding for this work.

## Abstract

### Background

Coronavirus disease 2019 or COVID-19 is caused by a newly discovered coronavirus, SARS-CoV-2. The Malaysian government has planned to procure COVID-19 vaccine through multiple agencies and companies in order to vaccinate at least 70% of the population. This study aimed to determine the knowledge, acceptance and perception of Malaysian adults regarding the COVID-19 vaccine.

### Methodology

An online survey was conducted for two weeks in December 2020. A bilingual, semi-structured questionnaire was set up using Google Forms and the generated link was shared on social media (i.e., Facebook and WhatsApp). The questionnaire consisted of questions on knowledge, acceptance and perception of COVID-19 vaccine. The association between demographic factors with scores on knowledge about COVID-19 vaccine were analysed using the Mann-Whitney test for two categorical variables, and the Kruskal-Wallis test used for more than two categorical variables.

### Results

A total of 1406 respondents participated, with the mean age of 37.07 years (SD = 16.05) years, and among them 926 (65.9%) were female. Sixty two percent of respondents had poor knowledge about COVID-19 vaccine (mean knowledge score 4.65; SD = 2.32) and 64.5% were willing to get a COVID-19 vaccine. High knowledge scores associated with higher education background, higher-income category and living with who is at higher risk of getting severe COVID-19. They were more likely to be willing to get vaccinated if they were in a lower age group, have higher education levels and were female.

### Conclusion

Even though knowledge about vaccine COVID-19 is inadequate, the majority of the respondents were willing to get vaccinated. This finding can help the Ministry of Health plan for

**Competing interests:** The authors have declared that no competing interests exist.

future efforts to increase vaccine uptake that may eventually lead to herd immunity against COVID-19.

## Introduction

Coronavirus disease 2019 or COVID-19 is caused by a newly discovered coronavirus, SARS-CoV-2. This new infection was believed to have emerged from Wuhan City, Hubei Province, China in December 2019. On March 11 2020, the World Health Organization (WHO) declared COVID-19 as a pandemic [1]. Until early June 2021, this emergent disease has infected more than 170 million people around the world and caused more than 3 million deaths [1]. The rate of infection had not seem to slow down in the majority of the affected countries, and varying degrees of lockdowns have been issued in the effort to contain the spread of the virus. In Malaysia, a resurgence of infections began in late September 2020 with a rapid increase in the number of infections, at more than 4000 cases daily since mid-January 2021 [2]. As of June 15 2021, Malaysia has reported 3968 COVID-19 deaths or a 0.6 percent fatality rate out of 662,457 cases [2].

Currently, there are more than 100 candidates of COVID-19 vaccines under development [3]. About 11 months after the emergence of the disease, the Food and Drug Administration (FDA) has approved the use of Pfizer/BioNTech and Moderna COVID-19 vaccines in a mass immunization programme [4]. Phase three clinical trials for Pfizer/BioNTech vaccines enrolled 43,661 participants, while Moderna vaccines involving 30,000 participants [5, 6]. The clinical trial results showed that these vaccines can protect recipients from a COVID-19 infection by forming antibodies and providing immunity against a COVID-19 virus [4]. There are also other companies in the race for vaccine development and in the final stages of trials. It is expected that many vaccines will be ready for distribution by early or mid-2021 [7]. The United Kingdom was among the first countries that have started mass immunization COVID-19 vaccine [8]. Apart from Moderna and Pfizer that use mRNA as the active substance, other vaccines use various other types of antigen such as viral vector, attenuated virus and inactivated virus [9]. The use of mRNA is a new technology for vaccine development, where the vaccine contains messenger RNA instructs cells to produce a protein that acts as an antigen.

As safe and effective vaccines are being made available, the next challenge will be dealing with vaccine hesitancy. Vaccine hesitancy, identified as one of the ten most important current health threats, is defined as the reluctance or refusal to vaccinate despite the availability of vaccines [10]. Wong et al. (2011) conducted a population-based study in Hong Kong on the acceptance of the COVID-19 vaccine using the health belief model (HBM) and found that perceived severity, perceived vaccine benefits, cues to action, self-reported health outcomes, and trust were all positive indicators of acceptance. Perceived vulnerability to infection had no significant association with acceptance, whereas perceived access barriers and harm were negative predictors [11]. In addition, another community-based study found that people's desire to get vaccinated against COVID-19 has fallen dramatically during the pandemic, with over half of the population were hesitant or unwilling to get vaccinated [12].

Misinformation and unsubstantiated rumours regarding COVID-19 vaccines have been around and repeatedly shared on social media platforms even before the release of an effective vaccine [13]. The use of mRNA genetic material in several vaccines have been sensationalized by some, with the false claims that the vaccine can alter human DNA [14] Additionally, the rapid development of COVID-19 vaccines has reportedly raised concerns regarding the safety

and long term effects, even among the medical staffs [15]. Findings from studies among healthcare workers (HCWs) are alarming, as a small percentage of HCWs do not intend to get the COVID-19 vaccine [16, 17].

The Malaysian government has procured COVID-19 vaccine through a government-to-government deal with the Republic of China, direct purchase from pharmaceutical companies and the COVID-19 Global Vaccine Access (Covax) Facility. With these arrangements, Malaysia is expected to receive its first batch of COVID-19 vaccines to immunise 6.4 million people as early as end of February 2021 [18]. We embarked on this study to determine the knowledge, acceptance and perception of the COVID-19 vaccine among the Malaysian adult population. The findings from this study will provide data and crucial information for the government to find strategies to increase public understanding and the uptake of COVID-19 vaccine.

## Methodology

This cross-sectional, online population-based survey was conducted from 1st to 15th December 2020. The study sample size was estimated using the Raosoft sample size calculator. A minimum of 385 participants were required at a margin of error of 5%, a 95% confidence interval (CI), and a population size of 32.6 million at a 50% response distribution. A bilingual, semi-structured questionnaire was adopted and adapted from Reiter et al. (2020) [19], and then set up via Google Forms. The access link was then shared via online platforms including Facebook and WhatsApp, initiated by all project members. The sharing was escalated by our family members, friends, colleagues, and acquaintances. The inclusion criteria for respondents' eligibility include those more than 18 years old, and an understanding of the Malay or English language. The respondents were requested to take part in the survey by completing the questionnaire without any time restrictions. Reliability measurement was tested earlier on 50 respondents for both the English and Malay version of the questionnaire. Cronbach alpha values for knowledge, perceived susceptibility, perceived barriers and perceived benefits were 0.718, 0.714, 0.714 and 0.834, respectively for the English version. Whereas the Cronbach alpha values for the Malay version were 0.665, 0.688, 0.787 and 0.889, respectively.

The questionnaire consists of four sections: Section A on demographic and COVID-19 status, Section B on the knowledge on COVID-19 vaccine, Section C on the acceptance of COVID-19 vaccine and Section D on perception based on the Health Belief Model (HBM). For section B (knowledge), participants were given three options: Yes, No and Do not know. One mark was given for any correct answer and 0 mark for any wrong answer and do not know answers. The maximum knowledge score was 10, and those who obtained marks above the median of the total score (6 and above) will be categorized as having good knowledge. Section C consists of questions on the willingness to take the vaccine and the reason, cost of the vaccine and factors influencing the decision. For Section D, five options were given: strongly agree, agree, neutral, disagree and strongly disagree, for perceived susceptibility and barriers. The questionnaire used in this study is not published under a CC-BY license, and other researchers may cite the related article when referencing the questionnaire.

### Ethical consideration

This research was approved by the Ethics Committee of Universiti Sains Islam Malaysia with the code project of USIM/JKEP/2021-126. The subjects consented to participate in this survey by volunteering to complete and submit the questionnaire.

### Study variables

**Dependent variables.** COVID-19 knowledge score.

Acceptance to COVID-19 vaccine.

**Independent variables.** Age, gender, educational status, income category, presence of any chronic diseases, history of been infected with COVID-19, history of family members or friends been infected with COVID-19, living with someone who is at higher risks of getting severe COVID-19 including living with elderly or family members with comorbidity or having long-term medical follow up or chronic medication.

## Data analysis

All data were entered into the Microsoft Excel spreadsheet and then loaded and coded into the SPSS version 23 software for final analysis. Simple descriptive analyses, including frequencies, percentages, mean, and standard deviation (SD) were computed for demographic characteristics, the knowledge scores regarding COVID-19 vaccine, and the perceived susceptibility, barriers and benefits to the COVID-19 vaccine. Histogram with normality curve and Kolmogorov–Smirnov test was used to check for the normal distribution of data in this study. Since the data were not normally distributed, the non-parametric tests were used for inferential analysis. The association between demographic factors with scores on knowledge regarding COVID-19 vaccine was analysed using the Mann-Whitney test for two categorical variables, and the Kruskal-Wallis test used for more than two categorical variables. A Chi-square test was carried out to determine the significant level of association and the relationship between the categorical independent variables of demographic factors and outcome variables of acceptance to the COVID-19 vaccine. Statistical significance was defined at $p < 0.05$.

## Results

### Demographic data

A total of 1406 respondents participated in this online survey. The mean age was 37.07 years (SD = 16.05; range = 18–81) and 926 (65.9%) of the respondents were female. The detailed characteristics of the respondents are shown in Table 1.

### Knowledge regarding COVID-19 vaccine

A total of 872 (62.0%) of the respondents had poor knowledge about COVID-19 vaccine (Fig 1). The statement "COVID-19 vaccines will be given via injection", had the most percentage of correct answers (82.1%). The statement with the lowest percentage of correct answers was "Everyone including children can receive COVID-19 vaccination" and "COVID-19 vaccine can also protect us from influenza," in which only 14.7% and 18.5% of respondents gave the correct answer. Table 2 shows the knowledge questions and scores for each statement.

Table 3 shows the association between demographic factors and knowledge scores. Higher education level, higher income and living with high-risk individuals were significantly associated with higher knowledge score.

### Acceptance towards COVID-19 vaccine

Almost two thirds of the respondents (64.5%) indicated willingness to get vaccinated (Fig 2). The majority agreed that the government should provide free vaccination to high-risk groups. More than 70% of the respondents would pay a maximum of RM 100 for the vaccine and only a small proportion (4.6%) reported not being able to afford the vaccine at any price. The effectiveness of the vaccine and suggestions from the Ministry of Health were the factors that most strongly influenced the decision to get the vaccination. Table 4 shows the details of the questions and their scores.

**Table 1. Socio-demographic characteristics (N = 1406).**

| No. | Characteristic | | *n* | % |
|---|---|---|---|---|
| 1. | Age | Mean: 37.07 (*SD* = 16.054)a | | |
| | | Median: 35 (*IQR* = 22–49) | | |
| | | Range: 18–91 | | |
| 2. | Age group | 18–29 | 602 | 42.8 |
| | | 30–39 | 214 | 15.2 |
| | | 40–49 | 259 | 18.4 |
| | | 50–59 | 156 | 11.1 |
| | | 60 and above | 175 | 12.4 |
| 3. Gender | | Male | 480 | 34.1 |
| | | Female | 926 | 65.9 |
| 4. Education | | No formal education | 2 | 0.1 |
| | | Secondary Education | 80 | 5.7 |
| | | Certificate or Diploma | 271 | 19.3 |
| | | Bachelor's Degree | 775 | 55.1 |
| | | Postgraduate studies (Master's or PhD) | 278 | 19.8 |
| 5. Income (RM) | | Mean: 11249.77 (*SD* = 29565.239) a [USD 2717 (SD = 7140)] | | |
| | | Median: 8000 (*IQR* = 4000–11000) a [USD 1932 (IQR = 966–2656)] | | |
| | | Range: 0–800000 [USD 0–193190] | | |
| 6. Income category* | | B40: Less than RM 4850 | 408 | 29.0 |
| | | M40: RM4851-RM10970 | 639 | 45.4 |
| | | T20: Above RM 10971 | 359 | 25.5 |
| 7. Chronic diseases | | Hypertension | 148 | 10.5 |
| | | Hypercholesterolemia | 134 | 9.5 |
| | | Diabetes mellitus | 93 | 6.6 |
| | | Chronic Lung Diseases | 52 | 3.7 |
| | | Heart Diseases | 36 | 2.6 |
| | | Cancer | 16 | 1.1 |
| | | Chronic Kidney Diseases | 4 | 0.3 |
| | | Others | 44 | 3.1 |

a Shapiro-Wilk p <0.001

*Department of Statistics, Malaysia, 2020. Household Income and Basic Amenities [20]

Table 5 shows the association of demographic factors and the acceptance to COVID-19 vaccines. Lower age group, higher education level, female, and not having chronic diseases were significantly associated with acceptance to COVID-19 vaccine.

## Perceived susceptibilities, barriers, benefits, and cues to action towards COVID-19 vaccine

About 55.9% perceived that they were able to spread the virus to other people and 30% of the respondents perceived that they were susceptible to get severe COVID-19 infection About 75% did not agree that COVID-19 vaccine could cause infection. More than half were worried about the vaccine's adverse effects and almost one third of them agreed that scary information about COVID-19 vaccine was rampant on social media. The majority believed that the vaccine could protect themselves and other people who are not vaccinated. Almost half were neutral in terms of vaccine cost and safety. Table 6 provides details of the perception scores. All

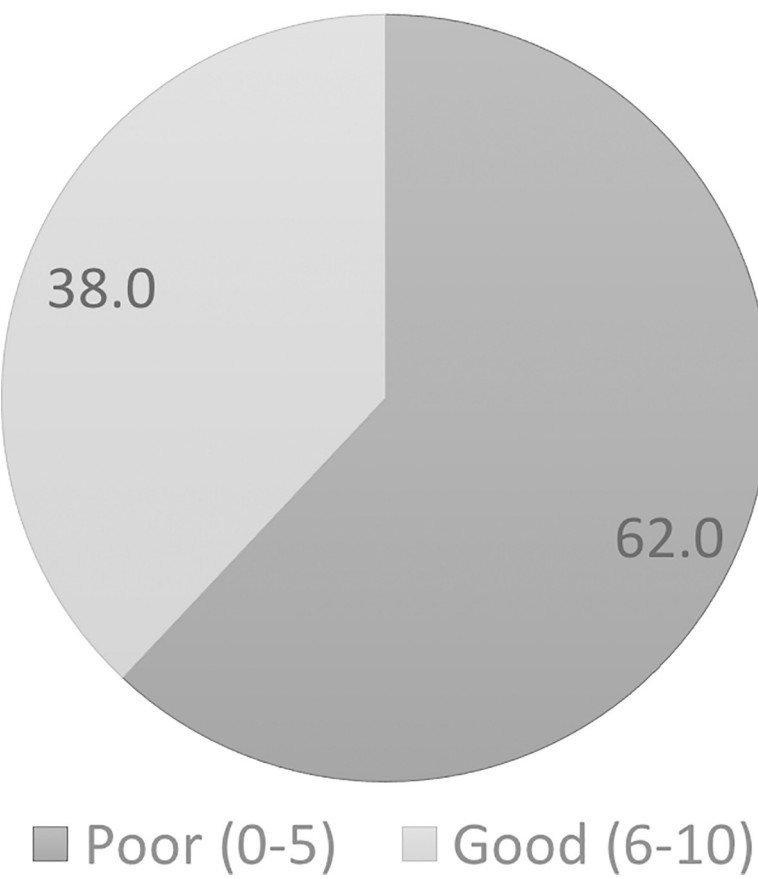

**Fig 1. Knowledge regarding COVID-19 vaccine category (N = 1406).**

components in HBM have a significant association with acceptance towards COVID-19, as shown in Table 7.

## Discussions

Since the announcement of the effectiveness of the two rapidly developed vaccines by Pfizer and Moderna, news and articles about vaccines have been circulating in the mass media and social media. This study found that electronic media and social media, including the Malaysian Ministry of Health (MOH) website were the most sought platforms for information regarding the COVID-19 vaccine. Only a small proportion of respondents received the information via newspapers, journal articles and medical-related websites. Previous studies have shown that the use of mass media can yield a positive impact on health-risk behaviours in the community [21]. However, with the recent advancement of informative technologies, social media is rapidly evolving and gaining more popularity than traditional mass media. Even the traditional mass media is adapting and evolving to fit into the social media platforms. Validated health information shared in social media delivers rapid and successful dissemination of knowledge [22]. On the other hand, excess information can lead to media fatigue, misinformation and the spread of fake news [23]. Health literacy is also an important aspect to determine the effectiveness of understanding and appraising the information [24].

To date, there is no published article about the level of knowledge of the COVID-19 vaccine among the Malaysian population. Previously, a study was done among Malaysian parents

**Table 2. Knowledge about COVID-19 vaccine (N = 1406).**

| No. | Statement | Mean (SD) | Correct | | Wrong/Do not Know | |
|---|---|---|---|---|---|---|
| | | | n | % | n | % |
| 1. COVID-19 vaccines use inactivated coronavirus as the antigen.* | | 0.40 (0.489) | 557 | 39.6 | 849 | 60.4 |
| 2.COVID-19 vaccines use genetic material from coronavirus as the active ingredient. * | | 0.37 (0.234) | 527 | 37.5 | 879 | 62.5 |
| 3. COVID-19 vaccine stimulates our body to produce antibody, T cells and memory cells to combat COVID-19 infection.* | | 0.73 (0.446) | 1021 | 72.6 | 385 | 27.4 |
| 4. COVID-19 vaccine protects the receiver from getting COVID-19 infection. * | | 0.74 (0.438) | 1043 | 74.2 | 363 | 25.8 |
| 5. COVID-19 vaccination may protect other people who do not receive vaccine. * | | 0.43 (0.495) | 598 | 42.5 | 808 | 57.5 |
| 6. Vaccine production involves animal study, 3 phases of clinical trials that cover thousands of people and evaluated by the authority to ensure the vaccine efficacy and safety. * | | 0.58 (0.493) | 820 | 58.3 | 586 | 41.7 |
| 7. COVID-19 vaccines will be given via injection. * | | 0.82 (0.384) | 1154 | 82.1 | 252 | 17.9 |
| 8. COVID-19 vaccines do not have side effects. | | 0.25 (0.431) | 346 | 24.6 | 1060 | 75.4 |
| 9. Everyone including children can receive COVID-19 vaccination. | | 0.46 (0.499) | 207 | 14.7 | 1199 | 85.3 |
| 10. COVID-19 vaccine can also protect us from influenza. | | 0.18 (0.388) | 260 | 18.5 | 1146 | 81.5 |

*Yes is the correct answer

revealed that poor knowledge has interfered with their decision on HPV vaccination among their children [25]. Inadequate knowledge regarding vaccination can be due to low education background, poor socioeconomic status or obtaining information from their peer layman [26, 27]. This study found that more than half of the respondents had poor knowledge of the COVID-19 vaccine. Higher education level, higher income and living with high-risk individuals were significantly associated with higher knowledge score.

Malaysian populations were found to have good knowledge, attitude and perception regarding COVID-19 prevention [28]. This is possibly the main reason for the higher acceptance of COVID-19 vaccine among the respondents, despite having low knowledge on the vaccine. Our acceptance rate is almost similar to Saudi Arabia (64.7%) [29] and the United Kingdom (64%) [30], better than Turkey (49.7%) [31], but lower than China (91.3%) [32] and Indonesia (93.3%) [33]. Lower age group, higher education level, female, and not having chronic diseases were significantly associated with acceptance to COVID-19 vaccine. In Saudi Arabia, willingness to accept the COVID-19 vaccine was relatively high among older age groups, married, education level postgraduate degree or higher, non-Saudi, and those employed in the government sector [29]. Although the acceptance rate is similar to Saudi Arabia, one distinct difference is that while in Malaysia, the younger age groups showed greater acceptance, in Saudi it is the older age groups.

The success of any vaccination programme to achieve herd immunity depends on the vaccine acceptance and uptake rate. The herd immunity threshold depends on the basic reproduction number (R0). With the R0 of 2–3, no population immunity and that all individuals are equally susceptible and equally infectious, the herd immunity threshold for SARS-CoV-2 would be expected to range between 50% and 67% in the absence of any interventions [34]. In this study, only one-third of the respondents were not willing to get vaccinated. This finding is in line with other studies done in other parts of the world. A study done in France in March

**Table 3. Association between demographic factors and knowledge score (N = 1406).**

| Variables | | Mean (*SD*) | Median (*IQR*) | *p*-value |
|---|---|---|---|---|
| Age group | 18–29 | 4.79 (2.167) | 5 (3) | .083a |
| | 30–39 | 4.60 (2.327) | 5 (3) | |
| | 40–49 | 4.86 (2.290) | 5 (4) | |
| | 50–59 | 4.25 (2.231) | 4 (4) | |
| | 60 and above | 4.65 (2.017) | 5 (3) | |
| Gender | Male | 4.63 (2.326) | 5 (3) | .786b |
| | Female | 4.66 (2.145) | 5 (3) | |
| Education | No formal education | 2.50 (3.536) | 2.5 (0) | < .001b* |
| | Secondary | 3.98 (2.444) | 4 (4) | |
| | Diploma | 4.30 (2.187) | 5 (3) | |
| | Degree | 4.70 (2.194) | 5 (3) | |
| | Master/PhD | 5.04 (2.100) | 5 (3) | |
| Income category | B40: Less than RM 4850 | 4.35 (2.285) | 4 (3) | < .001b* |
| | M40:RM4851-RM10970 | 4.62 (2.160) | 5 (3) | |
| | T20:Above RM 10971 | 5.03 (2.135) | 5 (3) | |
| Chronic diseases | Yes | 4.68 (2.207) | 5 (3) | .317b |
| | No | 4.53 (2.212) | 5 (3) | |
| Been infected with COVID-19 | Yes | 4.68 (2.207) | 5 (3) | .296b |
| | No | 4.53 (2.212) | 5 (3) | |
| Family members or friends been infected with COVID-19 | Yes | 4.65 (2.205) | 5 (3) | .815b |
| | No | 4.59 (2.234) | 5 (3) | |
| Live with someone who is at higher risk of getting severe COVID-19 | Yes | 4.65 (2.205) | 5 (3) | .003b* |
| | No | 4.59 (2.234) | 5 (3) | |

a Kruskal-Wallis test

b Mann-Whitney test

* significant at p-value < .05

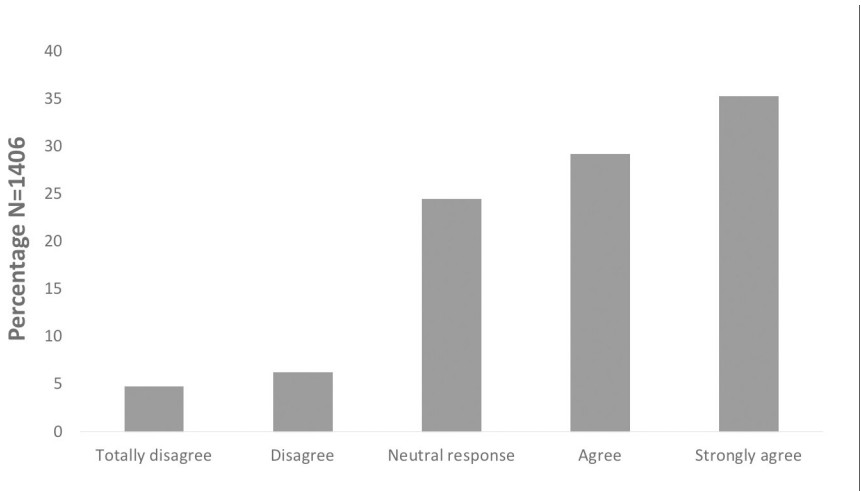

**Fig 2. Acceptance towards COVID-19 vaccine (N = 1406).**

**Table 4. Questions on factors influencing acceptance for COVID-19 vaccine (N = 1406).**

| No. | Statements | | *n* | % |
|---|---|---|---|---|
| 1. | In your opinion, should the government provide free COVID-19 vaccine to the high-risk groups? | Yes | 1272 | 90.5 |
| | | No | 30 | 2.1 |
| | | Do not know | 46 | 3.3 |
| | | Others | 58 | 4.1 |
| 2. | What is the most you would pay out of pocket to get the COVID-19 vaccine? | I can't afford to pay at all | 65 | 4.6 |
| | | less than RM50 [USD 12] | 550 | 39.1 |
| | | RM50 -RM100 [USD 12–24] | 484 | 34.4 |
| | | RM101-RM150 [USD 24–36] | 126 | 9.0 |
| | | More than RM150 [USD 36] | 46 | 3.3 |
| | | I don't mind any cost | 128 | 9.1 |
| | | Others | 7 | 0.5 |
| 3. | What are the factors that influence your decision to take the COVID-19 vaccine? | Effectiveness | 1028 | 73.1 |
| | | Suggestion from doctors or Ministry of Health | 874 | 62.2 |
| | | Number of positive COVID-19 cases | 691 | 49.1 |
| | | Adverse effects | 564 | 40.1 |
| | *This question allows multiple responses | Number of deaths due to COVID-19 | 542 | 38.5 |
| | | Health status | 539 | 38.3 |
| | | Cost | 482 | 34.3 |
| | | Duration of protection | 470 | 33.4 |
| | | Age | 414 | 29.4 |
| | | Type of vaccine | 362 | 25.7 |
| | | Number of Vaccine doses | 313 | 22.3 |
| | | Country that produces the vaccine | 263 | 18.7 |
| | | Suggestion from friends or family members | 203 | 14.4 |
| | | Others | 43 | 3.1 |

2020 showed that 26% of respondents refused vaccination, more prevalent among low-income people, young women and people older than 75 years old [30]. Another study done in the USA among the general population found that only 21% of respondents were not willing to be vaccinated [19]. Reasons for vaccine refusal including but not limited to safety, effectiveness, costs and side effects.

In order to achieve herd immunity, the vaccine hesitancy issue should be addressed. The Ministry of Health Malaysia has scaled up the vaccine promotional programmes, particularly through social media and mass media. More dialogues and forums involving experts from the ministry and universities have frequently been aired in the television and Facebook Live. In the beginning of the mass vaccination programme, the media also highlighted the vaccination process of the top leaders to increase the public confidence. Misinformation about the negative effects of vaccine from irresponsible parties had been monitored closely by the government. In addition, an emergency law to tackle fake news related to COVID-19 was introduced in March 2021 with hefty fines and jail terms of up to six years [35].

Worryingly, we found that those with existing chronic diseases have significantly lower acceptance rates than those who were healthy. Patients with cardiovascular disease, hypertension, diabetes, congestive heart failure, chronic kidney disease and cancer have been shown to have a greater risk of mortality compared to patients with COVID-19 without these comorbidities [36]. Early vaccination for this population is critical to ensure their health and safety.

**Table 5. Association between demographic factors and acceptance to COVID-19 vaccine (N = 1406).**

| Variables | | Response, n (%) | | | | | p-value[a] |
|---|---|---|---|---|---|---|---|
| | | Strongly disagree | Disagree | Neutral | Agree | Strongly agree | |
| Age group | 18–29 | 25 (37.3) | 15 (16.9) | 130 (37.8) | 144 (35.1) | 288 (58.1) | . < .001* |
| | 30–39 | 11 (16.4) | 18 (20.2) | 41 (11.9) | 76 (18.5) | 68 (13.7) | |
| | 40–49 | 14 (20.9) | 19 (21.3) | 63 (18.3) | 84 (20.5) | 79 (15.9) | |
| | 50–59 | 6 (9.0) | 11 (12.4) | 45 (13.1) | 53 (12.9) | 41 (8.3) | |
| | 60 and above | 11 (16.4) | 26 (29.2) | 65 (18.9) | 53 (12.9) | 20 (2.0) | |
| Gender | Male | 34 (50.7) | 28 (31.5) | 112 (32.6) | 130 (31.7) | 176 (35.5) | .035* |
| | Female | 33 (49.3) | 61 (68.5) | 232 (67.4) | 280 (68.3) | 320 (64.5) | |
| Education | No formal education | 0 | 0 | 2 (0.6) | 0 | 0 | . 022* |
| | Secondary Education | 6 (9.0) | 8 (9.0) | 14 (4.1) | 21 (5.1) | 31 (6.3) | |
| | Certificate or Diploma | 4 (6.0) | 15 (16.9) | 68 (19.8) | 87 (21.2) | 97 (19.6) | |
| | Bachelor's degree | 37 (55.2) | 36 (40.4) | 188 (54.7) | 233 (56.8) | 281 (56.7) | |
| | Postgraduate studies | 20 (29.9) | 30 (33.7) | 72 (20.9) | 69 (16.8) | 87 (17.5) | |
| Income category [#] | B40 | 20 (29.9) | 21 (23.6) | 102 (29.7) | 109 (26.6) | 156 (31.5) | .356 |
| | M40 | 26 (38.8) | 46 (51.7) | 165 (48.0) | 184 (44.9) | 218 (44.0) | |
| | T20 | 21 (31.3) | 22 (24.7) | 77 (22.4) | 117 (28.5) | 122 (24.6) | |
| Chronic diseases | Yes | 11 (16.4) | 31 (34.8) | 94 (27.3) | 99 (24.1) | 81 (16.4) | < .001* |
| | No | 56 (83.6) | 58 (65.2) | 250 (72.7) | 311 (75.9) | 414 (83.6) | |
| Been infected with COVID-19 | Yes | 0 | 1 (1.1) | 2 (0.6) | 1 (0.2) | 0 | .158 |
| | No | 67 (100) | 88 (98.9) | 342 (99.4) | 409 (99.8) | 496 (100) | |
| Familymembers or friends been infected with COVID-19 | Yes | 5 (7.5) | 12 (13.5) | 37 (10.8) | 41 (10.0) | 45 (9.1) | .667 |
| | No | 62 (92.5) | 77 (86.5) | 307 (89.2) | 369 (90.0) | 451 (90.9) | |
| Live with someone who is at higher risk of getting severe COVID-19 | Yes | 23 (34.3) | 31 (34.8) | 107 (31.1) | 119 (29.0) | 169 (34.1) | .515 |
| | No | 44 (65.7) | 58 (65.2) | 237 (68.9) | 291 (71.0) | 327 (65.9) | |

a Chi-square test

*p < .05

# Department of Statistics, Malaysia, 2020. Household Income and Basic Amenities [20]

Therefore, they will be given high priority for the COVID-19 vaccine. More information should be conveyed to this population about the risk of severe COVID-19 and the benefit of vaccination.

While the majority agreed that the government should provide free vaccination to the high-risk groups, more than 70% of the respondents would pay a maximum of RM 100 for the vaccine and only a small proportion (4.6%) cannot afford the vaccine. This is consistent with a previous Malaysian study in April 2020 when the COVID-19 vaccine was still in its early development [37]. However, the cost is not an issue since the Malaysian Government has decided to

**Table 6. Perception on susceptibilities, severity, barriers and benefits, and cues to action (N = 1406).**

| No. | Statements | Strongl y agree, n (%) | Agree, n (%) | Neutral, n (%) | Disagree, n (%) | Strongly disagree, n (%) |
|---|---|---|---|---|---|---|
| **Perceived Susceptibilities** | | | | | | |
| | I can spread the virus to other people | 226 (16.1) | 560 (39.8) | 355 (25.2) | 151 (10.7) | 114 (8.1) |
| **Perceived Severity** | | | | | | |
| | I am at risk of getting a severe COVID-19 infection. | 88 (6.3) | 335 (23.8) | 514 (36.6) | 310 (22.0) | 159 (11.3) |
| **Perceived Barriers** | | | | | | |
| | 1. COVID-19 vaccine may cause infection | 45 (3.2) | 180 (12.8) | 636 (45.2) | 636 (45.2) | 418 (29.7) |
| | 2. COVID-19 vaccine may not be effective | 52 (3.7) | 422 (30.0) | 645 (45.9) | 236 (16.8) | 51 (3.6) |
| | 3. I am worried about the adverse effects of the vaccine | 170 (12.1) | 579 (41.2) | 476 (33.9) | 140 (10.0) | 41 (2.9) |
| | 4. I am not sure whether or not I have to get the vaccine | 55 (3.9) | 302 (21.5) | 466 (33.1) | 389 (27.7) | 194 (13.8) |
| | 5. I don't have time to get the vaccine | 16 (1.1) | 46 (3.3) | 327 (23.3) | 620 (44.1) | 397 (28.2) |
| | 6. I don't have money to buy the vaccine | 34 (2.4) | 126 (9.0) | 554 (39.4) | 481 (34.2) | 211 (15.0) |
| | 7. Scary information about COVID-19 vaccines are rampant on social media | 86 (6.1) | 351 (25.0) | 481 (34.2) | 309 (22.0) | 179 (12.7) |
| | 8. It will be difficult to get vaccine from nearby clinic due to high demand | 118 (8.4) | 432 (30.7) | 512 (36.4) | 251 (17.9) | 93 (6.6) |
| **Perceived Benefits** | | | | | | |
| | 1. Vaccine protects me from getting infected | 396 (28.2) | 740 (52.6) | 219 (15.6) | 25 (1.8) | 26 (1.8) |
| | 2. Vaccine also protects other people who are vaccinated not | 384 (27.3) | 654 (46.5) | 239 (17.0) | 90 (6.4) | 39 (2.8) |
| | 3. After vaccination, I can lead a normal lifestyle | 281 (20.0) | 559 (39.8) | 422 (30.0) | 111 (7.9) | 33 (2.3) |
| Cues to action | | | | | | |
| | 1. Affordable cost | 121 (8.6) | 366 (26.0) | 787 (56.0) | 94 (6.7) | 38 (2.7) |
| | 2. Safe | 213 (15.1) | 506 (36.0) | 581 (41.3) | 77 (5.5) | 29 (2.1) |
| | 3. It is recommended by doctors and MOH | 330 (23.5) | 685 (48.7) | 336 (23.9) | 32 (2.3) | 23 (1.6) |
| | 4. Good information about vaccine in the mass media | 260 (18.5) | 561 (39.9) | 441 (31.4) | 97 (6.9) | 47 (3.3) |

give free vaccination to Malaysian citizen. The effectiveness of the vaccine and recommendation from the MOH were the highest factors that influence the decision to get the vaccination. This agrees with a study in Indonesia where 93.3% of respondents would like to be vaccinated if the vaccine is 95% effective and the acceptance decreased to 60.7% for a vaccine with 50% effectiveness [33].

A previous study done in Malaysia showed an increase in the perception of susceptibility to infection as the COVID-19 pandemic progressed [38]. Effective preventative behaviours such as personal hygiene and physical distancing to control SAR- CoV-2 transmission largely depend on the perceived susceptibility to infection [39, 40]. Perception of disease susceptibility also correlates with better health-seeking behaviour [41, 42].

**Table 7. Association between perception on susceptibilities, severity, barriers and benefits, and cues to action with acceptance towards COVID-19 vaccine (N = 1406).**

| Perception theme | Statements | | Acceptance, n (%) | | | | | p-value |
|---|---|---|---|---|---|---|---|---|
| | | | Strongly disagree | Disagree | Neutral | Agree | Strongly agree | |
| Perceived Susceptibilities | 1. I can spread the virus to other people | Strongly disagree | 13 (19.4) | 10 (11.2) | 22 (6.4) | 34 (8.3) | 35 (7.1) | < .001* |
| | | Disagree | 8 (11.9) | 12 (13.5) | 50 (14.5) | 45 (11.0) | 36 (7.3) | |
| | | Neutral | 20 (29.9) | 25 (28.1) | 116 (33.7) | 93 (22.7) | 101 (20.4) | |
| | | Agree | 19 (28.4) | 33 (37.1) | 119 (34.6) | 195 (47.6) | 194 (39.1) | |
| | | Strongly agree | 7 (10.4) | 9 (10.1) | 37 (10.8) | 43 (10.5) | 130 (26.2) | |
| Perceived Severity | 2. I am at risk of getting a severe COVID-19 infection. | Strongly disagree | 17 (25.4) | 14 (15.7) | 42 (12.2) | 33 (8.0) | 53 (10.7) | < .001* |
| | | Disagree | 11 (16.4) | 25 (28.1) | 82 (23.8) | 93 (22.7) | 99 (20.0) | |
| | | Neutral | 24 (35.8) | 26 (29.2) | 148 (43.0) | 155 (37.8) | 161 (32.5) | |
| | | Agree | 12 (17.9) | 21 (23.6) | 60 (17.4) | 113 (27.6) | 129 (26.0) | |
| | | Strongly agree | 3 (4.5) | 3 (3.4) | 12 (3.5) | 16 (3.9) | 54 (10.9) | |
| Perceived Barriers | 3. COVID-19 vaccine may cause infection | Strongly disagree | 13 (19.4) | 1 (1.1) | 11 (13.2) | 22 (5.4) | 80 (16.1) | < .001* |
| | | Disagree | 11 (16.4) | 10 (11.2) | 58 (16.9) | 144 (35.1) | 195 (39.3) | |
| | | Neutral | 21 (31.3) | 39 (43.8) | 224 (65.1) | 188 (45.9) | 164 (33.1) | |
| | | Agree | 12 (17.9) | 32 (36.0) | 40 (11.6) | 51 (12.4) | 45 (9.1) | |
| | | Strongly agree | 10 (14.9) | 7 (7.9) | 11 (3.2) | 5 (1.2) | 12 (2.4) | |
| | 4. COVID-19 vaccine may not be effective | Strongly disagree | 8 (11.9) | 1 (1.1) | 3 (0.9) | 5 (1.2) | 34 (6.9) | < .001* |
| | | Disagree | 7 (10.4) | 6 (6.7) | 22 (6.4) | 71 (17.3) | 130 (26.2) | |
| | | Neutral | 18 (26.9) | 17 (19.1) | 178 (51.7) | 207 (50.5) | 225 (45.4) | |
| | | Agree | 13 (19.4) | 55 (61.8) | 127 (36.9) | 127 (31.0) | 100 (20.2) | |
| | | Strongly agree | 21 (31.3) | 10 (11.2) | 14 (4.1) | 0 | 7 (1.4) | |
| | 3. I am worried about the adverse effects of the vaccine | Strongly disagree | 3 (4.5) | 0 | 3 (0.9) | 6 (1.5) | 29 (5.8) | < .001* |
| | | Disagree | 6 (9.0) | 0 | 10 (2.9) | 32 (7.8) | 92 (18.5) | |
| | | Neutral | 16 (23.9) | 11 (12.4) | 102 (29.7) | 151 (36.8) | 196 (39.5) | |
| | | Agree | 17 (25.4) | 42 (47.2) | 154 (44.8) | 205 (50.0) | 161 (32.5) | |
| | | Strongly agree | 25 (37.3) | 36 (7.9) | 75 (21.8) | 16 (3.9) | 18 (3.6) | |
| | 4. I am not sure whether or not I have to get the vaccine | Strongly disagree | 20 (29.9) | 4 (4.5) | 5 (1.5) | 22 (5.4) | 143 (28.8) | < .001* |
| | | Disagree | 13 (19.4) | 9 (9.0) | 30 (8.7) | 144 (35.1) | 194 (39.1) | |
| | | Neutral | 17 (25.4) | 24 (27.0) | 159 (46.2) | 160 (39.0) | 106 (21.4) | |
| | | Agree | 10 (14.9) | 37 (41.6) | 129 (37.5) | 80 (19.5) | 46 (9.3) | |
| | | Strongly agree | 7 (10.4) | 16 (8.0) | 21 (6.1) | 4 (1.0) | 7 (1.4) | |
| | 5. I don't have time to get the vaccine | Strongly disagree | 21 (31.3) | 16 (18.0) | 47 (13.7) | 92 (22.4) | 221 (44.6) | < .001* |
| | | Disagree | 18 (26.9) | 29 (32.6) | 151 (43.9) | 223 (54.4) | 199 (40.1) | |
| | | Neutral | 19 (28.4) | 33 (37.1) | 132 (38.4) | 77 (18.8) | 66 (13.3) | |
| | | Agree | 3 (4.5) | 7 (7.9) | 13 (3.8) | 15 (3.7) | 8 (1.6) | |
| | | Strongly agree | 6 (9.0) | 4 (4.5) | 1 (0.3) | 3 (0.7) | 2 (0.4) | |
| | 6. I don't have money to buy the vaccine | Strongly disagree | 15 (22.4) | 13 (14.6) | 34 (9.9) | 42 (10.2) | 107 (21.6) | < .001* |
| | | Disagree | 21 (31.3) | 32 (36.0) | 86 (25.0) | 160 (39.0) | 182 (36.7) | |
| | | Neutral | 21 (31.3) | 26 (29.2) | 177 (51.5) | 167 (40.7) | 163 (32.9) | |
| | | Agree | 6 (9.0) | 13 (14.6) | 39 (11.3) | 35 (8.5) | 33 (6.7) | |
| | | Strongly agree | 4 (6.0) | 5 (5.6) | 8 (2.3) | 6 (1.5) | 11 (2.2) | |
| | 7. Scary information about COVID -19 vaccines are rampant on social media | Strongly disagree | 7 (10.4) | 4 (4.5) | 13 (3.8) | 42 (10.2) | 113 (22.8) | < .001* |
| | | Disagree | 11 (16.4) | 14 (15.7) | 56 (16.3) | 100 (24.4) | 128 (25.8) | |
| | | Neutral | 18 (26.9) | 35 (39.3) | 144 (41.9) | 152 (37.1) | 132 (26.6) | |
| | | Agree | 18 (26.9) | 28 (31.5) | 110 (32.0) | 99 (24.1) | 96 (19.4) | |
| | | Strongly agree | 13 (19.4) | 8 (9.0) | 21 (6.1) | 17 (4.1) | 27 (5.4) | |
| | 8. It will be difficult to get vaccine from nearby clinic due to high demand | Strongly disagree | 10 (14.9) | 7 (7.9) | 6 (1.7) | 13 (3.2) | 57 (11.5) | < .001* |
| | | Disagree | 10 (14.9) | 19 (21.3) | 45 (13.1) | 79 (19.3) | 98 (19.8) | |
| | | Neutral | 25 (37.3) | 33 (37.1) | 158 (45.9) | 154 (37.6) | 142 (28.6) | |
| | | Agree | 15 (22.4) | 24 (27.0) | 107 (31.3) | 135 (32.9) | 151 (30.4) | |
| | | Strongly agree | 7 (10.4) | 6 (6.7) | 28 (8.1) | 29 (7.1) | 48 (9.7) | |

(*Continued*)

**Table 7.** (Continued)

| Perception theme | Statements | | Acceptance, n (%) | | | | | p-value |
|---|---|---|---|---|---|---|---|---|
| | | | Strongly disagree | Disagree | Neutral | Agree | Strongly agree | |
| Perceived Benefits | 1. Vaccine protects me from getting infected | Strongly disagree | 17 (25.4) | 3 (3.4) | 2 (0.6) | 2 (0.5) | 2 (0.4) | < .001* |
| | | Disagree | 7 (10.4) | 8 (9.0) | 4 (1.2) | 3 (0.7) | 3 (0.6) | |
| | | Neutral | 15 (22.4) | 42 (47.2) | 109 (31.7) | 41 (10.0) | 12 (2.4) | |
| | | Agree | 21 (31.3) | 33 (37.1) | 190 (55.2) | 289 (70.5) | 207 (41.7) | |
| | | Strongly agree | 7 (10.4) | 3 (3.4) | 39 (11.3) | 75 (18.3) | 272 (54.8) | |
| | 2. Vaccine also protects others people who are not vaccinated | Strongly disagree | 18 (26.9) | 3 (3.4) | 10 (2.9) | 4 (1.0) | 4 (0.8) | < .001* |
| | | Disagree | 10 (14.9) | 11 (12.4) | 30 (8.7) | 21 (5.1) | 18 (3.6) | |
| | | Neutral | 15 (22.4) | 33 (37.1) | 98 (28.5) | 55 (13.4) | 38 (7.7) | |
| | | Agree | 17 (25.4) | 36 (40.4) | 171 (49.7) | 257 (62.7) | 173 (34.9) | |
| | | Strongly agree | 7 (10.4) | 6 (6.7) | 35 (10.2) | 73 (17.8) | 263 (53.0) | |
| | 3. After vaccination, I can lead a normal lifestyle | Strongly disagree | 15 (22.4) | 6 (6.7) | 4 (1.2) | 4 (1.0) | 4 (0.8) | < .001* |
| | | Disagree | 9 (13.4) | 24 (27.0) | 38 (11.0) | 25 (6.1) | 15 (3.0) | |
| | | Neutral | 23 (34.3) | 45 (50.6) | 166 (48.3) | 91 (22.2) | 97 (19.6) | |
| | | Agree | 15 (22.4) | 11 (12.4) | 109 (31.7) | 234 (57.1) | 190 (38.3) | |
| | | Strongly agree | 5 (7.5) | 3 (3.4) | 27 (7.8) | 56 (13.7) | 190 (38.3) | |
| Cues to action | | | | | | | | |
| | 1. Affordable cost | Strongly disagree | 12 (17.9) | 6 (6.7) | 11 (3.2) | 2 (0.5) | 7 (1.4) | < .001* |
| | | Disagree | 6 (9.0) | 16 (18.0) | 33 (9.6) | 22 (5.4) | 17 (3.4) | |
| | | Neutral | 35 (52.2) | 51 (57.3) | 220 (64.0) | 213 (52.0) | 268 (54.0) | |
| | | Agree | 11 (16.4) | 15 (16.9) | 70 (20.3) | 141 (34.4) | 129 (26.0) | |
| | | Strongly agree | 3 (4.5) | 1 (1.1) | 10 (2.9) | 32 (7.8) | 75 (15.1) | |
| | 2. Safe | Strongly disagree | 19 (28.4) | 5 (5.6) | 2 (0.6) | 1 (0.2) | 2 (0.4) | < .001* |
| | | Disagree | 8 (11.9) | 32 (36.0) | 25 (7.3) | 6 (1.5) | 6 (1.2) | |
| | | Neutral | 19 (28.4) | 40 (44.9) | 225 (65.4) | 171 (41.7) | 126 (25.4) | |
| | | Agree | 16 (23.9) | 9 (10.1) | 66 (19.2) | 195 (47.6) | 220 (44.4) | |
| | | Strongly agree | 5 (7.5) | 3 (3.4) | 26 (7.6) | 37 (9.0) | 142 (28.6) | |
| | 3. It is recommended by doctors and MOH | Strongly disagree | 10 (14.9) | 5 (5.6) | 2 (0.6) | 3 (3.7) | 3 (0.6) | < .001* |
| | | Disagree | 8 (11.9) | 8 (9.0) | 9 (2.6) | 5 (1.2) | 2 (0.4) | |
| | | Neutral | 25 (37.3) | 45 (50.6) | 151 (43.9) | 62 (15.1) | 53 (10.7) | |
| | | Agree | 15 (22.4) | 28 (31.5) | 159 (46.2) | 263 (64.1) | 220 (44.4) | |
| | | Strongly agree | 9 (13.4) | 3 (3.4) | 23 (6.7) | 77 (18.8) | 218 (44.0) | |
| | 4. Good information about vaccine in the mass media | Strongly disagree | 14 (20.9) | 11 (12.4) | 10 (2.9) | 5 (1.2) | 7 (1.4) | < .001* |
| | | Disagree | 5 (7.5) | 16 (18.0) | 42 (12.2) | 18 (4.4) | 16 (3.2) | |
| | | Neutral | 25 (37.3) | 34 (38.2) | 160 (46.5) | 118 (28.8) | 104 (21.0) | |
| | | Agree | 14 (20.9) | 23 (25.8) | 111 (32.3) | 223 (54.4) | 190 (38.3) | |
| | | Strongly agree | 9 (13.4) | 5 (5.6) | 21 (6.1) | 46 (11.2) | 179 (36.1) | |

More than half of our respondents perceived that they could cause the spread of the virus. People with a higher perceived risk of COVID-19 infection are also more likely to support the vaccine [33]. The low percentage of perceived severity is likely due to the large number of younger respondents with no medical illness. Our respondents had perceived barriers to accepting the COVID-19 vaccine due to its adverse effects, vaccine availability and scary information about vaccines in social media. The majority perceived that vaccination could protect them and others from COVID-19 infection. This is consistent with other findings from other countries [43–45]. Moreover, the respondents believed that the vaccine is beneficial due to recommendations by the MOH and the fact that they can lead a normal life after vaccination. Conversely, more studies have to be done to assess the ability of vaccines to prevent disease transmissibility. According to the CDC, people should continue wearing masks, wash hands

frequently and practise physical distancing after getting the COVID-19 vaccine until herd immunity is achieved [46]. Perceived susceptibility, benefit and cues to action are associated with higher acceptance toward COVID-19 vaccine, and our finding is in concordance with other HBM studies [11, 47, 48].

To the best of our knowledge, this is the first study on the knowledge, acceptability and perception of COVID-19 vaccine in Malaysia. One limitation of this study was the use of convenience sampling via social media platforms. The distribution of the respondents might not reflect the actual population since most respondents were internet-savvy young adults. We suggest a larger study that includes respondents from diverse backgrounds, ethnicity, economic status and locations. Multiple public platform sharing is needed to increase the respondent's rate. Various data collection methods such as telephone interviews and face-to-face interviews should also be employed.

While waiting for vaccines to arrive, continuous education should be conducted to increase understanding and to clear up any misunderstandings or misinformation about the vaccine. Ideally, health education should be comprehensive and multilingual yet layman friendly. The important messages should reach out to all citizens from all walks of life, including those in the rural areas and technology illiterates. In addition to web-based and application-based educational tools, printed materials and face-to-face public talks may benefit certain groups of the population. Public talks involving religious groups can be conducted in the houses of worship by the experts.

## Conclusions

This study provides early insight into the Malaysian population's knowledge, acceptability and perception regarding COVID-19 vaccines. Knowledge about vaccines was relatively poor, particularly among low education levels, low income and not living with high-risk groups. The acceptability rate was significantly low among males, those with chronic diseases and those with low income. Education level of bachelor's degree and higher was associated with better acceptance towards COVID-19 vaccine. This finding can help the Ministry of Health to plan for future efforts to increase vaccine uptake that may eventually lead to herd immunity against SARS-CoV-2. The efforts should focus on those with insufficient knowledge and low acceptance, particularly those with chronic diseases and less financially fortunate people.

## Acknowledgments

We like to thank the Faculty of Medicine and Health Sciences and Universiti Sains Islam Malaysia for assisting us in publishing this paper.

## Author Contributions

**Conceptualization:** Nurul Azmawati Mohamed, Hana Maizuliana Solehan, Mohd Dzulkhairi Mohd Rani, Che Ilina Che Isahak.

**Data curation:** Nurul Azmawati Mohamed, Hana Maizuliana Solehan, Mohd Dzulkhairi Mohd Rani, Muslimah Ithnin.

**Formal analysis:** Nurul Azmawati Mohamed, Mohd Dzulkhairi Mohd Rani, Muslimah Ithnin.

**Funding acquisition:** Che Ilina Che Isahak.

**Investigation:** Nurul Azmawati Mohamed, Hana Maizuliana Solehan, Mohd Dzulkhairi Mohd Rani.

**Methodology:** Nurul Azmawati Mohamed, Hana Maizuliana Solehan, Mohd Dzulkhairi Mohd Rani, Muslimah Ithnin.

**Project administration:** Nurul Azmawati Mohamed, Hana Maizuliana Solehan, Mohd Dzulkhairi Mohd Rani.

**Software:** Mohd Dzulkhairi Mohd Rani.

**Supervision:** Che Ilina Che Isahak.

**Validation:** Nurul Azmawati Mohamed, Mohd Dzulkhairi Mohd Rani, Che Ilina Che Isahak.

**Visualization:** Nurul Azmawati Mohamed, Mohd Dzulkhairi Mohd Rani.

**Writing – original draft:** Nurul Azmawati Mohamed, Hana Maizuliana Solehan, Muslimah Ithnin.

**Writing – review & editing:** Nurul Azmawati Mohamed, Hana Maizuliana Solehan, Mohd Dzulkhairi Mohd Rani.

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
