## [Decision Letter · Decision Letter 0]

21 Apr 2021

PONE-D-21-00966

Knowledge, acceptance and perception on COVID-19 vaccine among Malaysians: A web-based survey

PLOS ONE

Dear Dr. Solehan,

Thank you for submitting your manuscript to PLOS ONE. After careful consideration, we feel that it has merit but does not fully meet PLOS ONE’s publication criteria as it currently stands. Therefore, we invite you to submit a revised version of the manuscript that addresses the points raised during the review process.

Your manuscript has some major issues which needs to be solved before considering it for publication. The methods section needs improvement as recommended by reviewers, consider improving discussion and language editing. you should clarify how the tool used in the study was generated and if it  was validated or not. full reviewers comments are enclosed.

We look forward to receiving your revised manuscript.

Kind regards,

Eman Sobh, M.D.

Academic Editor

PLOS ONE

Journal Requirements:

2. Please clarify in your Methods section whether the questionnaire is published under a CC-BY license, or whether you obtained permission from the publisher to reproduce the questionnaire in this manuscript. Please explain any copyright or restrictions on this questionnaire.

Reviewers' comments:

Reviewer's Responses to Questions

**Comments to the Author**

1. Is the manuscript technically sound, and do the data support the conclusions?

Reviewer #1: Partly

Reviewer #2: Yes

2. Has the statistical analysis been performed appropriately and rigorously? 

Reviewer #1: Yes

Reviewer #2: Yes

3. Have the authors made all data underlying the findings in their manuscript fully available?

Reviewer #1: Yes

Reviewer #2: Yes

4. Is the manuscript presented in an intelligible fashion and written in standard English?

Reviewer #1: Yes

Reviewer #2: Yes

5. Review Comments to the Author

Reviewer #1: Summary and general impression:

The paper made by N. A. Mohamed et al. team had great and fruitful efforts to discuss community knowledge, acceptance, and perception about the COVID-19 vaccines. It gave a good insight into the level of knowledge regarding nature, benefits, and risks among the Malaysian community as an important predictor of vaccine adoption and good effective community participation.

It also examined the acceptance proportion of the study units to the vaccination idea against COVID-19 emphasized the ability to pay for it and the common factors triggering for vaccination.

It also used the HBM as one of the popular models studying human behavior changes or health perceptions illustrating the benefits, barriers, cues to action, and other perspectives that could motivate or render the COVID-19 vaccination.

A large sample size of the study was good to validate the results putting the convenient sample technique limitations into consideration. Sociodemographic predictors of knowledge, acceptance, or perception are crucial, especially for recommendations directed to policymakers (who, where, what, when, and whom) to put in their priorities during COVID-19 vaccine health education and awareness campaigns planning.

Special issues:

Major issues:

1. The study should clarify the variables (dependent and independent) of the study in the methodology section to avoid confusion of the reader regarding the specific objectives of the study. It was hinted only at the last paragraph of the statistical analysis.

2. "Knowledge" is one of the dependent variables (outcomes) of the study it's categorized in the 2nd paragraph of the methodology to: good (≥6) and poor (<6) but its (table 3) relationship with Demographic factors weren’t presented according to that categorization (did not reveal who was good or poor regarding Demographic factors).

3. HBM has six fundamental perspectives (Benefits- Barriers- susceptibility- severity- Cues to action- self efficacy); (some other factors may be added by some psychiatrists in literature). So, Table 6 targets only three perspectives from them without any mentioned rationale or clarification even as a limitation. On the other hand, in the Susceptibilities section question (1) is severity perspective. Also, questions (3, 4, 6, and 7) in the benefits section are considered cues to action.

4. The HBM perspective association (relationship or prediction) with the acceptance of the COVID-19 vaccine is highly necessary to be presented in the results section to be able to recommend an appropriate situation analysis for active community mobilization intervention programmers.

Minor issues:

1. Title: "perception on" expression is not used a lot in this context; I think "perception about" is more informative.

2. Introduction: The authors should rewrite their Introduction to refer to the related literature of situation of the study outcomes worldwide, especially recently published work such as

• Wong MC, Wong EL, Huang J, Cheung AW, Law K, Chong MK, Ng RW, Lai CK, Boon SS, Lau JT, Chen Z. Acceptance of the COVID-19 vaccine based on the health belief model: A population-based survey in Hong Kong. Vaccine. 2021;39(7):1148–56.

• Shekhar R, sheik AB, Upadhyay S et al. COVID-19 Vaccine Acceptance among Health Care Workers in the United States. Vaccines (Basel). 2021;9(2):119. Published 2021 Feb 3. doi:10.3390/vaccines9020119

• Daly M, Robinson E. Willingness to Vaccinate against COVID-19 in the U.S.: Representative Longitudinal Evidence From April to October 2020 [published online ahead of print, 2021 Feb 15]. Am J Prev Med. 2021;doi:10.1016/j.amepre.2021.01.008

• Gagneux-Brunon A, Detoc M, Bruel S, et al. Intention to get vaccinations against COVID-19 in French healthcare workers during the first pandemic wave: a cross-sectional survey. J Hosp Infect. 2021;108:168-173. doi:10.1016/j.jhin.2020.11.020

3. Methodology: (page 5, 3rd sentence)

• The authors should clarify the reliability measurement was for the English version or Malay Version or both written together in the same form.

4. Data analysis: (page 6, 2nd sentence)

• Kolmogorov–Smirnov test is the standard for normality testing when the sample is more than 50.

5. Results:

• Table 2: score system analysis is considered a binary qualitative data (expressed 1: correct, 0: Not correct) which can't be presented as mean (SD) like continuous (scale) quantitative data; it will not be informative.

• Tables 1 and 5: Education categories' should follow the common international classifications or standardized (example: Diploma = High school, it may be conflicted with another postgraduate degree in other countries). Also, Currency should be converted (or symbol) to a dollar ($) to be more understood, especially with changes of currency all over the world.

• Table 5: The authors should clarify the operational definition (high-risk someone) in the variable "Live with someone who is at a higher risk of getting severe COVID- 19," what they meant?

Miscellaneous points:

The discussion section was written in a good and informative manner.

Reviewer #2: In this research survey from Malaysia, the authors attempt to better characterize public knowledge and acceptance regarding the SARS CoV 2 vaccine. Overall, this qualitative survey is reasonably well written. There are general grammatical errors throughout the document, and so a thorough proofreading will be needed prior to further submissions. I would also encourage the authors to review the manuscript (and, in particular, the introduction), to be more factual and succinct. I have other specific comments for the authors which are outlined below.

In the introduction, please refrain from using social media references (such as the BBC). In addition, please rework paragraph 2 on page 3 to more accurately reflect the number of people enrolled each vaccine trial, and that these vaccines were effective at preventing symptomatic/severe disease, and data on preventing disease transmission is less robust.

In the methodology, you mention intra-respondent consistency in knowledge/susceptibility, barriers, and benefits. I think review of these values and their implications would be better done in the results.

Your data analysis seems straightforward and is described reasonably well.

Table 4 is particularly interesting to me. It may be worth considering what roles the ministry of health or private doctors could provide in addressing vaccine hesitancy in the discussion.

Also, am I correct in assuming that those with a masters/PhD degree were less likely than those with a diploma/degree to be neutral or positive about acceptance of a COVID-19 degree? If so, why do you think that is (could also be addressed in the conclusions).

There needs to be a section that better outlines the limitations of this study. For instance, the demographics of your study group (in particular the gender makeup) are not consistent with World Bank estimates. In addition, those who might use Facebook or other social media platforms might have a different level of knowledge and acceptance of COVID-19 vaccines then the general population. It would be important to mention these limitations and how they may affect the conclusions.

6. PLOS authors have the option to publish the peer review history of their article (what does this mean?). If published, this will include your full peer review and any attached files.

Reviewer #1: **Yes: **Ahmed M. Yousef

Reviewer #2: No

---

## [Author Response · Author response to Decision Letter 0]

8 Jul 2021

Here is a point-by-point response to the reviewers’ comments and concerns.

Comments from Reviewer 1 

Major issues:

1. The study should clarify the variables (dependent and independent) of the study in the methodology section to avoid confusion of the reader regarding the specific objectives of the study. It was hinted only at the last paragraph of the statistical analysis.

Response: Thank you for pointing this out. To avoid confusion to the reader, we have added the variables subheading in the methodology section. 

Study Variables

Dependent Variables

COVID-19 knowledge score.

Acceptance to COVID-19 vaccine.

Independent Variables

Age, gender, Educational status, Income category, present of any Chronic diseases, history of been infected with COVID-19, history of family members or friends been infected with COVID-19, living with someone who is at higher risks of getting severe COVID-19 including living with elderly or family members with comorbidity or having long-term medical follow up or chronic medication.

2. "Knowledge" is one of the dependent variables (outcomes) of the study it's categorized in the 2nd paragraph of the methodology to: good (≥6) and poor (<6) but its (table 3) relationship with Demographic factors weren’t presented according to that categorization (did not reveal who was good or poor regarding Demographic factors).

Response: We thank the reviewer for this point. One of the primary rationales of the study is to investigate the association between demographic factors and knowledge of the respondents. In table 3, we are testing the association between the independent variables of demographic factors and total knowledge scores of the respondents. As the dependent variable, knowledge scores are continuous data, thus the mean and median of each independent variable is presented in Table 3.

3. HBM has six fundamental perspectives (Benefits- Barriers- susceptibility- severity- Cues to action- self efficacy); (some other factors may be added by some psychiatrists in literature). So, Table 6 targets only three perspectives from them without any mentioned rationale or clarification even as a limitation. On the other hand, in the Susceptibilities section question (1) is severity perspective. Also, questions (3, 4, 6, and 7) in the benefits section are considered cues to action.

Response: 

We thank the reviewer for this point. We have made the correction on the table 6 accordingly. In response to this, we have made amendment to methodology section, paragraph 2, last sentence.

 “For Section D, five options were given: strongly agree, agree, neutral, disagree and strongly disagree, for perceived susceptibility, perceived severity, perceived barriers, perceived benefits and cue to action.”

4. The HBM perspective association (relationship or prediction) with the acceptance of the COVID-19 vaccine is highly necessary to be presented in the results section to be able to recommend an appropriate situation analysis for active community mobilization intervention programmers.

Response: In corresponds to this point raised by reviewer, table 7 on the HBM perspective association with the acceptance of the COVID-19 vaccine is presented in the results section.

Minor issues:

1. Title: "perception on" expression is not used a lot in this context; I think "perception about" is more informative.

Response: Changed the word ‘on’ to ‘about’

2. Introduction: The authors should rewrite their Introduction to refer to the related literature of situation of the study outcomes worldwide, especially recently published work such as

• Wong MC, Wong EL, Huang J, Cheung AW, Law K, Chong MK, Ng RW, Lai CK, Boon SS, Lau JT, Chen Z. Acceptance of the COVID-19 vaccine based on the health belief model: A population-based survey in Hong Kong. Vaccine. 2021;39(7):1148–56.

• Shekhar R, sheik AB, Upadhyay S et al. COVID-19 Vaccine Acceptance among Health Care Workers in the United States. Vaccines (Basel). 2021;9(2):119. Published 2021 Feb 3. doi:10.3390/vaccines9020119

• Daly M, Robinson E. Willingness to Vaccinate against COVID-19 in the U.S.: Representative Longitudinal Evidence From April to October 2020 [published online ahead of print, 2021 Feb 15]. Am J Prev Med. 2021;doi:10.1016/j.amepre.2021.01.008

• Gagneux-Brunon A, Detoc M, Bruel S, et al. Intention to get vaccinations against COVID-19 in French healthcare workers during the first pandemic wave: a cross-sectional survey. J Hosp Infect. 2021;108:168-173. doi:10.1016/j.jhin.2020.11.020

Response: We appreciate the reviewer’s comment. We have accordingly added all the related literature as suggested by the reviewer in the introduction section.

As safe and effective vaccines are being made available, the next challenge will be in dealing with vaccine hesitancy. Vaccine hesitancy, identified as one of the ten most important current health threats, is defined as the reluctance or refusal to vaccinate despite the availability of vaccines.10 Wong et al. (2011) conducted a population-based study in Hong Kong on the acceptance of the COVID-19 vaccine using the health belief model (HBM) and found that perceived severity, perceived vaccine benefits, cues to action, self-reported health outcomes, and trust were all positive indicators of acceptance. Perceived vulnerability to infection had no significant association with acceptance, whereas perceived access barriers and harm were negative predictors. 11 In addition, another community-based study found that during the pandemic, people's desire to get vaccinated against COVID-19 has fallen dramatically, with over half of the population hesitant or unwilling to get vaccinated. 12

Misinformation and unsubstantiated rumors regarding COVID-19 vaccines have been around and repeatedly shared on social media platforms even before the release of an effective vaccine.13 The use of mRNA genetic material in several vaccines also have been sensationalized by some, with the false claims that the vaccine can alter human DNA. 14 Additionally, the rapid development of COVID-19 vaccines has reportedly raised concerns regarding the safety and long term effects, even among medical staff.15 Findings from studies among healthcare workers (HCWs) are especially concerning, as a small percentage of HCWs do not intend to get the COVID-19 vaccine. 16,17

3. Methodology: (page 5, 3rd sentence)

3.1 The authors should clarify the reliability measurement was for the English version or Malay Version or both written together in the same form.

Response: Thank you for pointing this out. The reliability assessment conducted through a pre-test survey to determine the Cronbach’s α of the questionnaire. Cronbach alpha for both English and Malay version has been added and reported separately in the methodology section.

Reliability measurement was tested earlier on 50 respondents for both the English and Malay version of the questionnaire. Cronbach alpha values for knowledge, perceived susceptibility, perceived barriers and perceived benefits were 0.718, 0.714, 0.714 and 0.834, respectively for the English version. Whereas the Cronbach alpha values for the Malay version were 0.665, 0.688, 0.787 and 0.889, respectively.

3.2 Data analysis: (page 6, 2nd sentence) Kolmogorov–Smirnov test is the standard for normality testing when the sample is more than 50.

Response: Thank you for pointing this out. We agree with this comment. In response to this point, we had made amendment to the sentence.

Histogram with normality curve and Kolmogorov–Smirnov test was used to check for the normal distribution of data in this study.

4 Results:

4.1 Table 2: score system analysis is considered a binary qualitative data (expressed 1: correct, 0: Not correct) which can't be presented as mean (SD) like continuous (scale) quantitative data; it will not be informative.

Response: Agree. We have, accordingly, modified the Table 2. The mean (SD) was removed from the Table 2.

4.2 Tables 1 and 5: Education categories' should follow the common international classifications or standardized (example: Diploma = High school, it may be conflicted with another postgraduate degree in other countries). Also, Currency should be converted (or symbol) to a dollar ($) to be more understood, especially with changes of currency all over the world.

Response: Thank you. As suggested, the education categories were amended following the common international classification for Tables, 1, 3 and 5. Information on the converted currency of dollar ($) were added next to the Ringgit Malaysia currency in Table 1 and 4.

4.3 Table 5: The authors should clarify the operational definition (high-risk someone) in the variable "Live with someone who is at a higher risk of getting severe COVID- 19," what they meant?

Response: We agree with this and have added the details on the definition used for "Live with someone who is at a higher risk of getting severe COVID- 19," in the methodology section, in the independent variables of variables subheading.

….. living with someone who is at higher risks of getting severe COVID-19 including living with elderly or family members with comorbidity or having long-term medical follow up or chronic medication.

Comments from Reviewer 2

1. In the introduction, please refrain from using social media references (such as the BBC). In addition, please rework paragraph 2 on page 3 to more accurately reflect the number of people enrolled each vaccine trial, and that these vaccines were effective at preventing symptomatic/severe disease, and data on preventing disease transmission is less robust.

Response: Thank for pointing this. As suggested, we had revised the reference used for paragraph 2 that used the social media reference. Additionally, we also have rework paragraph 2 reflect the number of people enrolled each vaccine trial, and that these vaccines were effective in protecting the recipient from a COVID-19 infection.

Currently, there are more than 100 candidates of COVID-19 vaccines under development.3 About 11 months after the emergence of the disease, the Food and Drug Administration (FDA) has approved the use of Pfizer/BioNTech and Moderna COVID-19 vaccines in a mass immunization programme.4 Phase three clinical trials for Pfizer/BioNTech vaccines enrolled 43,661 participants, while Moderna vaccines involving 30,000 participants. 5,6 The clinical trial results showed that these vaccines can protect recipients from a COVID-19 infection by forming antibodies and providing immunity against a COVID-19 virus. 4 There are also other companies in the race for vaccine development and in the final stages of trials. It is expected that many vaccines will be ready for distribution by early or mid-2021.7 The United Kingdom was among the first countries that have started mass immunization COVID-19 vaccine. 8 

8 Mathieu, E., Ritchie, H., Ortiz-Ospina, E. et al. A global database of COVID-19 vaccinations. Nat Hum Behav. 2021. https://doi.org/10.1038/s41562-021-01122-8

2. In the methodology, you mention intra-respondent consistency in knowledge/susceptibility, barriers, and benefits. I think review of these values and their implications would be better done in the results.

Response: We appreciate the reviewer's question. After discussion we would like to retain the intra-respondent consistency of the knowledge, susceptibility, barriers, and benefits of the questionnaire in the methodology section of the manuscript. The Cronbach’s alpha value reported in this section is a value of the internal consistency of the score from the pilot test. This value was commonly reported in the methodology section of research articles.

Reference: Taber, K.S. The Use of Cronbach’s Alpha When Developing and Reporting Research Instruments in Science Education. Res Sci Educ 48, 1273–1296 (2018). https://doi.org/10.1007/s11165-016-9602-2.

3. Table 4 is particularly interesting to me. It may be worth considering what roles the ministry of health or private doctors could provide in addressing vaccine hesitancy in the discussion.

Response: Thank you for pointing it out. We have added it in the discussion

4. Also, am I correct in assuming that those with a masters/PhD degree were less likely than those with a diploma/degree to be neutral or positive about acceptance of a COVID-19 degree? If so, why do you think that is (could also be addressed in the conclusions).

Response: Thank you for pointing this out. We have found that those with higher education 

level (i.e. bachelor’s degree and master/PhD) have positive acceptance towards COVID-19 

vaccine. We have included this in the conclusion.

5. There needs to be a section that better outlines the limitations of this study. For instance, the demographics of your study group (in particular the gender makeup) are not consistent with World Bank estimates. In addition, those who might use Facebook or other social media platforms might have a different level of knowledge and acceptance of COVID-19 vaccines then the general population. It would be important to mention these limitations and how they may affect the conclusions.

Response: Thank you for pointing this out. We have added the limitations in the discussion

---

## [Decision Letter · Decision Letter 1]

2 Aug 2021

Knowledge, acceptance and perception on COVID-19 vaccine among Malaysians: a web-based survey

PONE-D-21-00966R1

Dear Dr. Solehan,

We’re pleased to inform you that your manuscript has been judged scientifically suitable for publication and will be formally accepted for publication once it meets all outstanding technical requirements. Please note to add study data to the link  as per your data statement you provided (the link contains link to results and not to data).

Kind regards,

Eman Sobh, M.D.

Academic Editor

PLOS ONE

Additional Editor Comments (optional):

Reviewers' comments:

Reviewer's Responses to Questions

**Comments to the Author**

1. If the authors have adequately addressed your comments raised in a previous round of review and you feel that this manuscript is now acceptable for publication, you may indicate that here to bypass the “Comments to the Author” section, enter your conflict of interest statement in the “Confidential to Editor” section, and submit your "Accept" recommendation.

Reviewer #1: All comments have been addressed

Reviewer #2: All comments have been addressed

2. Is the manuscript technically sound, and do the data support the conclusions?

Reviewer #1: Yes

Reviewer #2: (No Response)

3. Has the statistical analysis been performed appropriately and rigorously? 

Reviewer #1: Yes

Reviewer #2: (No Response)

4. Have the authors made all data underlying the findings in their manuscript fully available?

Reviewer #1: Yes

Reviewer #2: (No Response)

5. Is the manuscript presented in an intelligible fashion and written in standard English?

Reviewer #1: Yes

Reviewer #2: (No Response)

6. Review Comments to the Author

Reviewer #1: (No Response)

Reviewer #2: (No Response)

7. PLOS authors have the option to publish the peer review history of their article (what does this mean?). If published, this will include your full peer review and any attached files.

Reviewer #1: No

Reviewer #2: No

---

## [Editor Report · Acceptance letter]

5 Aug 2021

PONE-D-21-00966R1 

Knowledge, acceptance and perception on COVID-19 vaccine among Malaysians: a web-based survey 

Dear Dr. Solehan:

I'm pleased to inform you that your manuscript has been deemed suitable for publication in PLOS ONE. Congratulations! Your manuscript is now with our production department. 

Kind regards, 

on behalf of

Dr. Eman Sobh 

Academic Editor

PLOS ONE